

# Extremely low genetic variability within and among locations of the greenfish holothurian *Stichopus chloronotus* Brandt, 1835 in Okinawa, Japan

Taha Soliman[1,2,3], Okuto Takama[1], Iria Fernandez-Silva[1,4,5] and James D. Reimer[1,6]

[1] Molecular Invertebrate Systematics and Ecology Laboratory, Graduate School of Engineering and Science, University of the Ryukyus, Nishihara, Okinawa, Japan
[2] National Institute of Oceanography and Fisheries, Alexandria, Egypt
[3] Microbiology and Biochemistry of Secondary Metabolites Unit, Okinawa Institute of Science and Technology Graduate University, Onna, Okinawa, Japan
[4] Section of Ichthyology, California Academy of Sciences, San Francisco, CA, USA
[5] Department of Biochemistry, Genetics and Immunology, Campus Universitario, University of Vigo, Vigo, Spain
[6] Tropical Biosphere Research Center, University of the Ryukyus, Nishihara, Okinawa, Japan

Corresponding author
Taha Soliman,
tahasoliman2000@yahoo.com

## ABSTRACT

The greenfish sea cucumber *Stichopus chloronotus* is an economically and ecologically important sea cucumber species throughout its range. This species is widely distributed, inhabiting coral reefs of the Indo-Pacific Ocean. Our study evaluated population genetic structure and levels of genetic diversity in southern Japan. A total of 180 individuals were collected from eight locations from Okinawa and Okinoerabu Islands and sequenced using mitochondrial 16S ribosomal DNA (16S) and nuclear histone H3 (H3) gene. Only three 16S haplotypes were detected (518 bp) with haplotype diversity ranging from 0 to 0.56 and nucleotide diversity from 0 to 0.1%. H3 showed no variation among the studied locations. It is plausible that such results could be due to a shift to asexual reproduction. Additionally, the presence of the species on the east coast of Okinawa could only be detected in one location and all individuals consisted of a single haplotype. Genetic differences between the east and west coasts of Okinawa have been noticed in other coral reef organisms, and attributed to either ecological or biogeographical historical differences between the coasts due to differing levels of isolation during Pleistocene ice ages. Results from the present study should inform management and conservation policies of *S. chloronotus* in southern Japan.

## INTRODUCTION

Sea cucumbers (holothurians) belong to class Holothuroidea of phylum Echinodermata and play an important and essential part in maintaining and structuring the marine ecosystem. They are deposit feeders of carbonate sand and rubble through their digestive tracts and dissolve $CaCO_3$ as part of their digestive process (*Schneider et al., 2011*). As

'ecosystem engineers,' healthy populations of sea cucumbers are important for maintaining the continuing health of many marine ecosystems.

However, sea cucumbers of the order Aspidochirotida (families Holothuroidea and Stichopodidae) are suffering from overexploitation and overfishing in several countries (*Anderson et al., 2011*; *Friedman et al., 2011*; *Purcell, 2010*; *Toral-Granda, Lovatelli & Vasconcellos, 2008*) due to high demand from Chinese and Southeast Asian markets where they are consumed as food and utilized for traditional medicine (*Bordbar, Anwar & Saari, 2011*). Unfortunately, statistics on fisheries/fishing activities for sea cucumbers are unavailable for many countries as most fisheries organizations still record invertebrate catch as 'others.' However, some organizations such as the Secretariat of the Pacific Community (SPC) and the Food and Agriculture Organization of the United Nations (FAO) have made efforts to make records available for invertebrates including sea cucumbers.

The species *Stichopus chloronotus* Brandt, 1835 (greenfish sea cucumber), belonging to the family Stichopodidae, is a member of such highly commercial marine sea cucumber fisheries. This species is a widely distributed across most of the tropical Indo-Pacific region, from the Western Indian Ocean to the Central Pacific, and from southern Japan to northern Australia (*Purcell et al., 2012*). This species inhabits shallow waters from the intertidal zone to depths of 15 m and can be found on coral reef flats and slopes in dense numbers. *Conand, Uthicke & Hoareau (2002)* reported on the sex ratios of *S. chloronotus* in some areas of the Indian and Pacific Oceans and found a very high number of males compared to females. *S chloronotus* is known to reproduce by two methods; by sexual reproduction through indirect development (pelagic auricularia larvae) and asexually by transverse fission (*Uthicke, 1997*; *Uthicke, 1999*; *Uthicke, 2001*). In some marine invertebrates the pelagic larval stage, produced by sexual reproduction, has a large and long-term effect on gene flow via pelagic larval dispersal (*Ellingson & Krug, 2016*; *Pechenik, 1999*). The duration of the pelagic larval stage (PLD) of *S. chloronotus* is not well known, although it may be approximately 20 days as seen in other Stichopodidae (*Soliman et al., 2013*). In the northwestern Pacific, larval dispersal of *S. chloronotus* is influenced by the Kuroshio Current, which is a major ocean current bringing tropical water north from tropical regions along of the Ryukyu Archipelago and Japan's southeast Pacific coast (*Barkley, 1970*).

In Okinawa, Japan, similar to many other regions of the world, sea cucumbers have been under increasing commercial pressure due to high demand. A recent study showed that some populations of *Holothuria edulis* Lesson, 1830 around Okinawa Island had lower than expected genetic diversity, indicating potential overharvesting and/or ecosystem degradation (*Soliman, Fernandez-Silva & Reimer, 2016*). Prefectural regulations have been put in place to try and limit further damage to sea cucumber populations, but basic genetic and abundance data are lacking for all other sea cucumber species for the entire Ryukyu Archipelago, and for Okinawa Island in particular. In order to properly conserve sea cucumbers around areas of large-scale human populations and coastal development like Okinawa Island, genetic population data are urgently needed.

Population genetic structure studies of *S. chloronotus* on the Great Barrier Reef have previously been carried out through polymorphic allozyme electrophoresis (*Uthicke, Benzie & Ballment, 1999*) and amplified fragment length polymorphism (AFLP) markers (*Uthicke*

**Table 1** Molecular diversity of Stichopus chloronotus of mitochondrial 16S ribosomal DNA (16S) sequences from eight locations (= populations) in southern Japan (*n*: Sample size; *h*: number of haplotypes; $h_d$: haplotype diversity; $\pi$: nucleotide diversity).

| Locations | Lat. and Long. | *n* | *h* | $h_d$ | $\pi$ | Fu's Fs | Fu's Fs *p*-value |
|---|---|---|---|---|---|---|---|
| Okinoerabu Island | 27°25″05.2″N 128°39″00.6″E | 24 | 1 | 0.0000 | 0.0000 | 0.0000 | N.A. |
| Yona, Okinawa Island | 26°45″56.6″N 128°11″42.4″E | 24 | 1 | 0.0000 | 0.0000 | 0.0000 | N.A. |
| Motobu, Okinawa Island | 26°40″45.8″N 127°52″55.4″E | 24 | 2 | 0.0833 | 0.0002 | −1.0279 | 0.079 |
| Ryugu, Okinawa Island | 26°31″48.6″N 127°55″34.8″E | 21 | 2 | 0.5143 | 0.0011 | 1.4737 | 0.757 |
| Zanpa, Okinawa Island | 26°25″58.8″N 127°42″57.2″E | 21 | 1 | 0.0000 | 0.0000 | 0.0000 | N.A. |
| Oyama, Okinawa Island | 26°17″01.2″N 127°44″20.5″E | 22 | 3 | 0.5584 | 0.0013 | 0.1995 | 0.504 |
| Itoman, Okinawa Island | 26°07″15.1″N 127°39″34.7″E | 24 | 2 | 0.1594 | 0.0003 | −0.2489 | 0.212 |
| Teniya, Okinawa Island | 26°33″52.6″N 128°08″16.0″E | 15 | 1 | 0.0000 | 0.0000 | 0.0000 | N.A. |

*& Conand, 2005*). In addition, there has been one study on the characterization of microsatellite markers for *S. chloronotus* utilizing specimens from the Ryukyu Islands, Japan, although there was no discussion of biogeographical findings (*Taquet et al., 2011*). The allozyme and AFLP studies suggested that population structure and genetic diversity of *S. chloronotus* is affected by asexual reproduction, with lower than expected genetic diversity due to the presence of clones.

In the present study, we assess the genetic diversity and population structure among eight *S. chloronotus* locations in southern Japan (Okinawa and Okinoerabu Islands) using sequences of mitochondrial DNA (16S ribosomal DNA) and nuclear DNA (histone H3), in order to provide genetic connectivity. We examine whether *S. chloronotus* on the east and west coasts of Okinawa are genetically differentiated as has been observed in a number of recent studies on other sea cucumber (*Soliman, Fernandez-Silva & Reimer, 2016*) and other invertebrate species (*White, Reimer & Lorion, 2016*). This information will be useful for future conservation and fisheries management. Based on recent increasing commercial pressure in Okinawa, we also expect that locations near to large human populations may show reduced *S. chloronotus* genetic diversity, as well as a potential shift to asexual reproduction due to reduced water quality, as was previously observed in *H. edulis* (*Soliman, Fernandez-Silva & Reimer, 2016*).

## MATERIALS AND METHODS

A total of 180 specimens of the greenfish sea cucumber *S chloronotus* were collected from February to April 2015 by snorkeling and SCUBA diving at seven locations around Okinawa Island (Itoman, Oyama, Zanpa, Ryugu, Motobu, Yona, Teniya) and one location at Okinoerabu Island, Kagoshima, Japan (Table 1 and Fig. 1). During our surveys, *S. chloronotus* was not found at any locations on the eastern coast of Okinawa Island asides from at Teniya (T Soliman et al., 2015, unpublished data). Samples of *S. chloronotus* were obtained non-lethally through body wall-tissue biopsy (1 cm²) as in our previous study (*Soliman, Fernandez-Silva & Reimer, 2016*) and preserved in absolute ethanol in the field. Live animals were released back to the same location from where they were collected.

Total genomic DNA was extracted from a small piece (0.25 g) of body wall tissue using a DNeasy Blood & Tissue extraction kit (Qiagen, Tokyo, Japan) following the manufacturer's
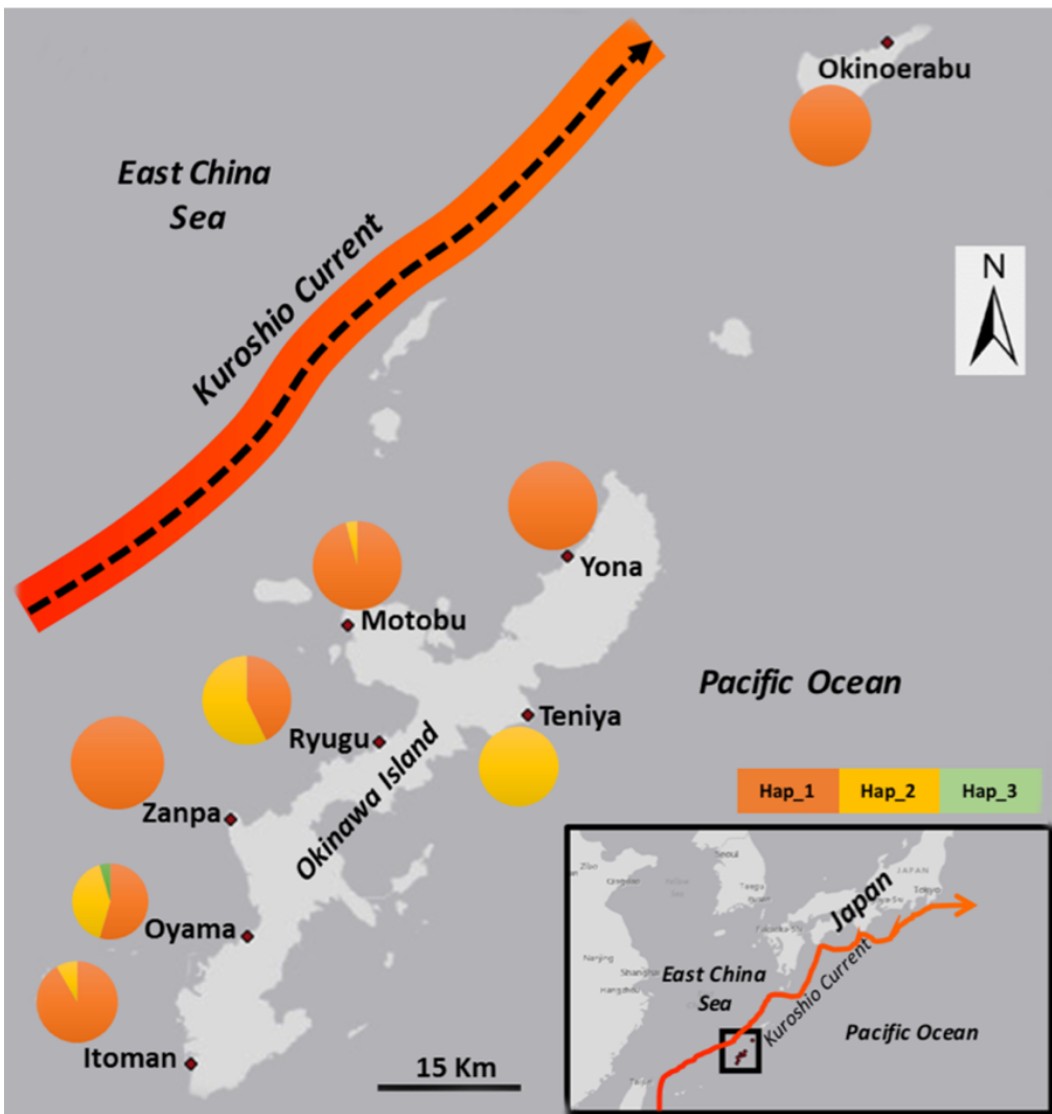

**Figure 1** **Map with sampling locations and frequency of 16S ribosomal DNA (16S) haplotypes of Stichopus chloronotus from Okinawa and Okinoerabu Islands, Japan.** The arrows depict the prevailing path and direction of the Kuroshio Current..

protocol. Polymerase chain reactions (PCR) were used to amplify fragments of two DNA markers; mitochondrial 16S and nuclear H3 gene, using the following primers: 16SA-R 5′-CGC CTG TTT ATC AAA AAC AT-3′; 16SB-R 5′-GCC GGT CTG AAC TCA GAT CAC GT-3′ (*Palumbi et al., 1991*), and H3a: 5′-ATG GCT CGT ACC AAG CAG ACV GC-3′; H3b: 5′-ATA TCC TTR GGC ATR ATR GTG AC-3′ (*Colgan et al., 1998*), respectively. Amplification reactions were carried out in a 15 μl volume containing 5 to 20 ng of template genomic DNA, 0.5 μM of each primer, and 10 μl of HotStarTaqTM Master Mix (Qiagen, Tokyo, Japan), in deionized water. PCR reactions were performed on an Eppendorf thermocycler (Eppendorf AG, Hamburg, Germany) with an initial denaturation step at 95 °C for 15 min, 35 cycles of denaturation at 94 °C for 1 min, annealing at 46 °C (16S) and

48 °C (H3) for 1 min and extension at 72 °C for 1 min, and a final extension step at 72 °C for 10 min. The amplification products were tested using electrophoresis (1.5% agarose gel) and purified with Exonuclease I and Alkaline Phosphatase Shrimp (Takara) by incubation at 37 °C for 20 min, followed by deactivation at 83 °C for 30 min. Purified PCR products were sequenced using an ABI Prism automated sequencer at Fasmac Co., Kanagawa, Japan (http://www.fasmac.co.jp/index.html), in both forward and reverse directions.

Consensus sequences of 16S and H3 were aligned and edited using Geneious v.8.1.3 (http://www.geneious.com, *Kearse et al., 2012*), and MEGA v.6.0 (*Tamura et al., 2013*). The H3 sequences showed no variation and no additional analyses were conducted. For 16S, number of haplotypes ($h$), haplotype diversity ($h_d$) and nucleotide diversity ($\pi$) were estimated with DNASP v.5.10.01 (*Librado & Rozas, 2009*). The best-fit model of DNA sequence evolution was estimated using MEGA v.6.0 (*Tamura et al., 2013*). The AIC (Akaike Information Criterion) indicated that Jukes-Cantor (JC) was the best-fit model. We computed pairwise $\Phi_{ST}$ and average of pairwise differences (Nei's distance, within and between locations) for all pairs of locations using Arlequin 3.5.1.2 (*Excoffier & Lischer, 2010*; *Weir & Cockerham, 1984*). The simultaneous tests correction was adjusted using the modified false discovery rate (FDR) method (*Narum, 2006*). To investigate genetic partitioning among regions, among locations within regions, and between all locations we applied a hierarchical analysis of molecular variance (AMOVA), using Arlequin, and tested the following hypothesis; A—Three groups: Okinoerabu Island as group 1, Teniya (east) as group 2 and all other western locations (Itoman, Motobu, Oyama, Ryugu, Yona, Zanpa) as group 3; and B—Two groups: Teniya (east) as group 1 and western locations of Okinawa Island (Itoman, Motobu, Oyama, Ryugu, Yona, Zanpa) plus Okinoerabu Island as group 2. Principal Coordinates Analysis (PCoA) based on 16S haplotype frequencies among locations were conducted and visualized using PAST 3.11 (*Hammer, Harper & Ryan, 2001*).

## RESULTS

We obtained sequences of a 518 bp fragment of 16S from 175 specimens (GenBank Accession numbers KX522950–KX523124) from eight total locations around Okinawa and Okinoerabu Islands (Table 1 and Fig. 1). The alignment of 16S sequences resulted in three haplotypes: Hap_1 was common among seven locations (Itoman, Oyama, Zanpa, Ryugu, Motobu, Yona, Okinoerabu), Hap_2 was shared among five locations (Itoman, Oyama, Ryugu, Motobu, Teniya), and Hap_3 was exclusive to Oyama (Figs. 1 and 2). Four locations showed no haplotype diversity for 16S; all 69 sea cucumbers sampled in Zanpa, Yona (both on Okinawa's west coast), and Okinoerabu were Hap_1 while all 15 specimens sampled at Teniya (Okinawa's east coast) consisted of Hap_2 (Table 1 and Fig. 1). Overall, the haplotype diversity was low ($h_d = 0.36$), and was highest in the Oyama ($h_d = 0.558$) and Ryugu ($h_d = 0.5143$) locations. In addition, the nucleotide diversity of 16S was low for all locations combined ($\pi = 0.0008$; Table 1).

We also obtained a 307 bp fragment of the nuclear H3 gene from 180 individuals of *S. chloronotus* collected in the same locations around Okinawa and Okinoerabu Islands. Obtained H3 sequences showed no variation. Therefore all subsequent population genetics analyses were based on 16S data only.

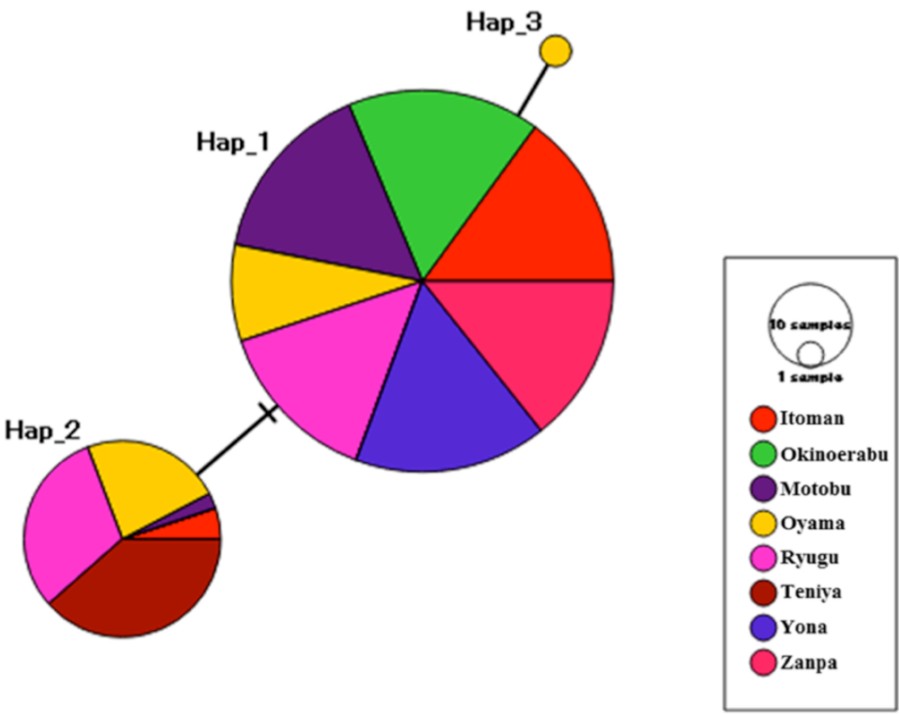

**Figure 2  Median-joining haplotype network inferred from 16S ribosomal DNA sequences of *Stichopus chloronotus.*** Each circle represents a different haplotype and the size of a circle is proportional to the frequency of each haplotype. Colours denote the proportion of haplotypes collected at each location as indicated by the embedded key. Haplotypes separated by black crossbars differ by a single nucleotide, whereas hatch marks indicate unsampled haplotypes.

Significant pairwise $\Phi_{ST}$ values were found in 17 out of 28 comparisons after multiple tests correction ($P < 0.01$) (Table 2 and Fig. 3). Most of the significant comparisons were between locations not sharing haplotypes, Teniya (east coast of Okinawa Island) vs. Yona and Zanpa (west coast of Okinawa Island) or with Okinoerabu Island. Although the null hypothesis of panmixia in the overall AMOVA test (global $\Phi_{ST} = 0.559$, $P < 0.001$) was rejected, results of the hierarchical AMOVAs did not support significant differences between east and west coasts or between north (Okinoerabu Island), east of Okinawa Island (Teniya) and west of Okinawa Island (Itoman, Motobu, Oyama, Ryugu, Yona, Zanpa) (Table 3 and Fig. 3).

Principle Coordinates Analyses (PCoA) using overall pairwise $\Phi_{ST}$ for 16S revealed three clusters; (1) Yona, Zanpa, Motobu, Itoman and Okinoerabu, (2) Ryugu and Oyama, and (3) Teniya (Fig. 4).

## DISCUSSION

The present study revealed low levels of genetic diversity of the greenfish sea cucumber within the southern Ryukyu Islands of Japan, with only three haplotypes in 16S and one in H3. These low levels of polymorphism in *S. chloronatus* impose some caution in the overall interpretation of results. Various studies of *S. chloronotus* across regions using

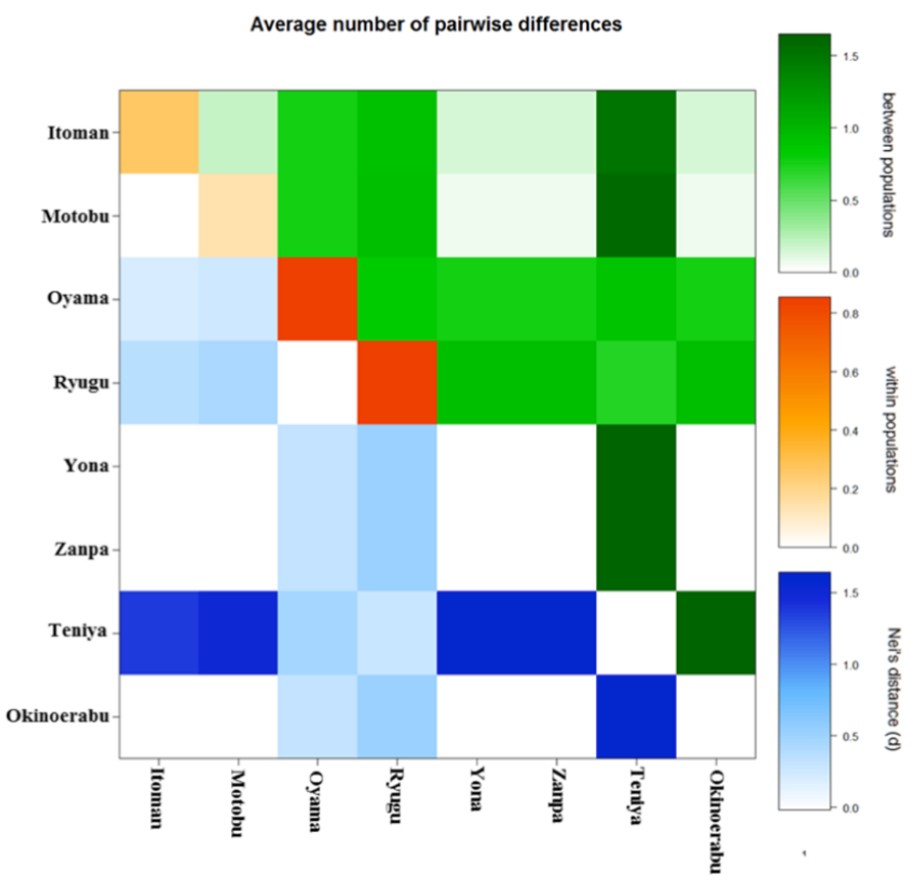

**Figure 3** Average number of pairwise differences within and among populations of Stichopus chloronotus for 16S ribosomal DNA.

**Table 2** Pairwise estimates of $\Phi_{ST}$ among eight locations of *Stichopus chloronotus* in southern Japan inferred from a 518 bp DNA fragment of the mitochondrial 16S ribosomal gene from 175 specimens.

| Location | Okinoerabu | Yona | Motobu | Ryugu | Zanpa | Oyama | Itoman | Teniya |
|---|---|---|---|---|---|---|---|---|
| Okinoerabu | 0.000 | | | | | | | |
| Yona | 0.000 | 0.000 | | | | | | |
| Motobu | 0.020 | 0.000 | 0.000 | | | | | |
| Ryugu | **0.569** | **0.569** | **0.486** | 0.000 | | | | |
| Zanpa | 0.000 | 0.000 | 0.006 | **0.550** | 0.000 | | | |
| Oyama | **0.441** | **0.441** | **0.351** | 0.021 | **0.422** | 0.000 | | |
| Itoman | 0.044 | 0.044 | 0.028 | **0.408** | 0.036 | **0.271** | 0.000 | |
| Teniya | **1.000** | **1.000** | **0.947** | **0.355** | **1.000** | **0.473** | **0.894** | 0.000 |

**Notes.**

Significant $\Phi_{ST}$ at $P < 0.01$ following the False Discovery Rate (FDR) correction (Narum, 2006) indicated in bold.

different genetic markers have also revealed low genetic diversity. *Uthicke & Conand (2005)* observed low genetic diversity in the southwestern Pacific (Great Barrier Reef and Torres Strait) compared to the western Indian Ocean using amplified fragment length polymorphism (AFLP) analyses, and suggested that AFLP markers are a suitable method

**Table 3** Hierarchical analyses of molecular variance (AMOVA) based a 518 bp fragment of the mitochondrial 16S ribosomal gene from 175 specimens of *S. chloronotus* testing different hypotheses of geographical groupings among eight locations in southern Japan. Hypothesis A: Group 1, Okinoerabu Island; Group 2, Teniya; Group 3, Itoman, Motobu, Oyama, Ryugu, Yona, Zanpa. Hypothesis B: Group 1, Okinoerabu, Itoman, Motobu, Oyama, Ryugu, Yona, Zanpa; Group 2, Teniya.

| Variation | d.f. | SS | Variation (%) | Fixation indices | P value | d.f. | SS | Variation (%) | Fixation indices | P value |
|---|---|---|---|---|---|---|---|---|---|---|
| **Hypothesis** | | | | **A** | | | | | **B** | |
| Among groups | 2 | 17.22 | 47.08 | $\Phi_{CT} = 0.4781$ | 0.1691 | 1 | 16.09 | 70.85 | $\Phi_{CT} = 0.7085$ | 0.1202 |
| Among population within groups | 5 | 11.56 | 20.28 | $\Phi_{SC} = 0.3885$ | <0.0001 | 6 | 12.7 | 11.56 | $\Phi_{SC} = 0.3966$ | <0.0001 |
| Within populations | 167 | 22.06 | 31.91 | $\Phi_{ST} = 0.6782$ | <0.0001 | 167 | 22.06 | 17.59 | $\Phi_{ST} = 0.8241$ | <0.0001 |

**Notes.**

d.f., degree of freedom; SS, sum of squares.

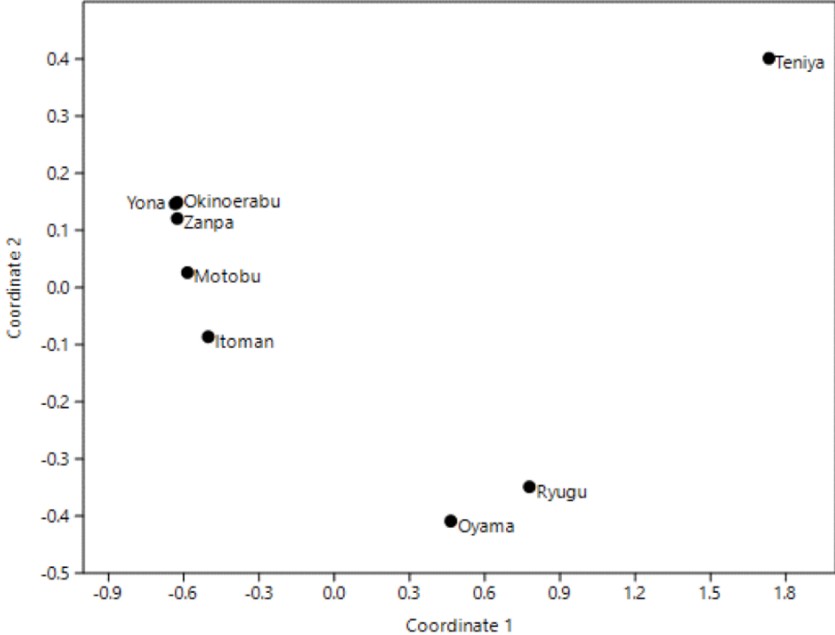

**Figure 4** Principal Coordinates Analysis (PCoA) based on 16S ribosomal DNA haplotype frequencies of Stichopus chloronotus populations from Okinawa and Okinoerabu Islands.

to examine clonality in sea cucumbers. In addition, allozyme studies have showed a low number of polymorphic loci in *S. chloronotus* in some locations such as the Great Barrier Reef, Torres Strait, and La Reunion (*Uthicke, Benzie & Ballment, 1999*; *Uthicke, Conand & Benzie, 2001*). The low genetic diversity results may stem from reported high rates of asexual reproduction in *S. chloronotus* in several other Indo-Pacific regions, as well as a biased male sex ratio (*Conand, Uthicke & Hoareau, 2002*). In addition, strong variation of rates of asexual reproduction over small geographic scales in *S. chloronotus* has been reported in genetic surveys and reproductive biology studies (*Conand, Uthicke & Hoareau,*

*2002*; *Uthicke, Benzie & Ballment, 1999*). This pattern seems to be common in other sea cucumbers: low genetic diversity has been observed in *Holothuria edulis* around Okinawa Island using mitochondrial (16S and COI) and nuclear H3 genes (*Soliman, Fernandez-Silva & Reimer, 2016*), and in *Holothuria atra* in Okinawa and the Ogasawara Islands based on COI sequences (*Skillings, Bird & Toonen, 2011*). We posit that our observations of low genetic diversity could be attributed to shifts to asexual reproduction in *S. chloronotus*.

Low habitat stability combined with high seawater concentrations of nutrients (eutrophication) and small individual sizes (caused by over-exploitation) have been identified as the most important factors that can lead to asexual reproduction in sea cucumbers (*Conand, Uthicke & Hoareau, 2002*). Over the past 40 years, coastal development and landfill have reduced and degraded shallow water habitats and water quality along the shoreline of Okinawa Island, and this has been implicated in reductions in biodiversity of near-shore marine communities (*Reimer et al., 2015*). These anthropogenic impacts may also at least partially explain the reduction in genetic diversity among these locations. Overharvesting can also lead to reductions of genetic diversity (*Smith, Francis & McVeagh, 1991*; *Pinsky & Palumbi, 2014*) and from our field observations at different sites of Okinawa Island it appears that some sea cucumber species such as *H. edulis* and *S. chloronotus* are overexploited (*Soliman, Fernandez-Silva & Reimer, 2016*). Similar anecdotal evidence on a reduction in sea cucumber numbers led Okinawa Prefecture to implement a harvesting licensing system in 2013 with further strengthening in 2016.

The haplotype frequency differences between the west coast of Okinawa and the single location on the east coast may be due to the existence of habitat differences, oceanographic barriers to dispersal promoting a reduction of larval exchange, and/or historical factors. We extensively searched for *S. chloronatus* individuals on the east coast, however, Teniya was the only location on this coast where the species could be found, further indicating the possibility of the above mentioned factors. Furthermore, our results indicated the presence of gene exchange among all Okinawa Island west coast locations and Okinoerabu Island. This pattern is consistent with the route of the Kuroshio Current flowing northward to the west of Okinawa Island (Fig. 1), and suggests this current may play a key role in connecting locations of *S. chloronatus*. On the other hand, areas isolated from the Kuroshio Current, such as the east coast of Okinawa Island, may harbor genetically distinct populations. Extensive geographic fine grid surveys on the east coast of Okinawa Island and from regions more isolated from the Kuroshio Current (e.g., the Daito Islands 310 km to the east of Okinawa Island; the disputed Senkaku Islands 410 km west of Okinawa Island) in southern Japan are needed to confirm this assertion. Similar patterns of genetic differentiation between east and west Okinawa Island locations have been observed in recent studies on the sea cucumber *Holothuria edulis* (*Soliman, Fernandez-Silva & Reimer, 2016*) and in the amphipod *Leucothoe vulgaris* (*White, Reimer & Lorion, 2016*). *White, Reimer & Lorion (2016)* suggested this reduction was caused by Pleistocene sea level changes (*Ni et al., 2014*) and/or geographic discontinuity (*Castelin et al., 2012*). In contrast, in one species of scleractinian coral (*Acropora digitifera*), Okinawa Island locations from both east and west coasts showed no separation and were treated as one panmictic population (*Shinzato et al., 2015*). Differences in the level of genetic structure could be explained by the shorter pelagic

larval duration in the sea cucumber (approx. 20 days in Stichopodidae; *Soliman et al., 2013*) than in *A. digitifera* (approx. 45 days; *Nishikawa & Sakai, 2005*). However, multiple studies point to the complexity of making predictions on genetic connectivity based on the length of pelagic larval development (e.g., *Selkoe et al., 2014*; *Shanks, 2009*). Nevertheless, from the few population genetics studies published in sea cucumbers to date (*Eastwood, Lopez & Drew, 2016*; *Skillings, Bird & Toonen, 2014*; *Soliman, Fernandez-Silva & Reimer, 2016*; *Valente, Serrao & Gonzalez-Wanguemert, 2015*; *Vergara-Chen et al., 2010*) there is a generalized pattern of highly geographically structured genetic variation and reduced dispersal ability, which may compromise the persistence of sea cucumber populations and species when facing intense fishing pressure, habitat degradation, or other environmental threats.

In conclusion, the present genetic survey of *S. chloronotus* indicates low levels of genetic diversity and reduced genetic exchange among sites in Okinawa and Okinoerabu islands, and particularly between the east and west coasts of Okinawa Island. These results will be useful for fisheries management decisions regarding this species in southern Japan. Both *Soliman, Fernandez-Silva & Reimer (2016)* and the present study also suggest that more research on sea cucumbers is needed from the Ryukyu Islands in order to properly conserve and protect the genetic diversity of these economic and ecologically important animals.

## ACKNOWLEDGEMENTS

S Hisashi is deeply thanked for his help on Okinoerabu Island—he will be deeply missed. We also thank R Diaz for help with sample collection and members of the MISE laboratory for their logistic and scientific support.

### Funding

TS was supported by Ministry of Higher Education of the Egyptian Government during this study in Japan. JDR was funded by a Japan Society for the Promotion of Science (JSPS) 'Zuno-Junkan' grant entitled 'Studies on origin and maintenance of marine biodiversity and systematic conservation planning.' IF-S was funded by a JSPS postdoctoral fellowship for overseas researchers and the European Union Seventh Framework Programme (FP7/2007-2011) under grant agreement PIOF-GA-2011-302957. The funders had no role in study design, data collection and analysis, decision to publish, or preparation of the manuscript.

### Grant Disclosures

The following grant information was disclosed by the authors:
Ministry of Higher Education of the Egyptian Government.
Japan Society for the Promotion of Science (JSPS).
JSPS postdoctoral fellowship for overseas researchers.
European Union Seventh Framework Programme: FP7/2007-2011, PIOF-GA-2011-302957.

## Competing Interests

James Reimer is an Academic Editor for PeerJ.

## Author Contributions

- Taha Soliman conceived and designed the experiments, performed the experiments, analyzed the data, wrote the paper, prepared figures and/or tables, reviewed drafts of the paper, sampling.
- Okuto Takama conceived and designed the experiments, analyzed the data, reviewed drafts of the paper, sampling.
- Iria Fernandez-Silva conceived and designed the experiments, analyzed the data, wrote the paper, reviewed drafts of the paper.
- James D. Reimer conceived and designed the experiments, analyzed the data, contributed reagents/materials/analysis tools, wrote the paper, prepared figures and/or tables, reviewed drafts of the paper.

## DNA Deposition

The following information was supplied regarding the deposition of DNA sequences:
GenBank Accession numbers KX522950–KX523124.

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
