# Peer review of "Extremely low genetic variability within and among locations of the greenfish holothurian Stichopus chloronotus Brandt, 1835 in Okinawa, Japan"

_PeerJ, doi:10.7717/peerj.2410_

## Round 0.1 · original submission · Minor Revisions

· Academic Editor

Minor Revisions

Thank you for submitting your manuscript to PEERJ. First my apologies for the late response: most of the reviewers were unresponsive or declined, which delayed the assessment of your article.

With that said, after careful consideration, I feel that your work has merit, but is not suitable for publication as it currently stands. Therefore, my decision is "Minor Revision.”

I am concerned particularly with comments from Reviewer 1, which are not fully addressed in the paper, namely the lack of heterozygotes in the individuals surveyed. This is a very important aspect that needs to be addressed extensively.

I encourage you to read the reviews and the attached pdf files with referees notes carefully and respond to each and every individual comment. If you are willing to send a thoroughly carefully revised version, I would be happy to reconsider your manuscript resubmission.

·

Basic reporting

Good

Experimental design

Good

Validity of the findings

There is a very odd result where not 1 in 200 sea cukes surveyed was a heterozygote at a nuclear locus- all homozygotes. The authors invoke that asexual reproduction could explain this, but asexual reproduction is not associated with homozygosity at the H3 locus. Further, these species are diploid.

There should be heterozygotes, unless...

The best explanation of the data is that there are different groups of reproductively-isolated sea cukes that are being surveyed - ie cryptic species. In each reproductively isolated group, the H3 locus is invariant (which would be consistent with a wide variety of marine inverts). If this is not the case, then the next best explanation is that the method employed to sequence H3 could not detect heterozygotes. I believe that the former is much more likely given the excellent work that these researchers have done in the past.

Given the conservation implications, I strongly suggest that the authors at the very least discuss the possibility that this species of sea cuke is composed of a cryptic species assemblage.

Additional comments

see attached

Reviewer 2 ·

Basic reporting

No Comments

Experimental design

no comments

Validity of the findings

no comments

Annotated reviews are not available for download in order to protect the identity of reviewers who chose to remain anonymous.

·

Basic reporting

Basic reporting is fine - pass.

Experimental design

Experimental design is generally ok, but see my comments below.

Validity of the findings

The raw sequences produced in this study should be deposited in a public database.
All of the alignments and input files for e.g., Arlequin, should be made available either as supplementary data or submitted to public repositories, such as Dryad or Figshare.

Additional comments

In this manuscript, Soliman et al. report on the genetic structure of a seacucumber in Okinawa (Japan), using partial mitochondrial 16S rDNA and nuclear histone H3 gene sequences as genetic markers. The manuscript uses standard and established methods and is in most parts clearly written, but I suggest, however, a few improvments that should be considered before the paper can be accepted.

L47-50: please enlighten the uninitiated reader what "important and essential part" holothurians exactly play in the marine ecosystem. This is a very broad and bold statement that needs more specificity and justification. Also, how do they do it?

L72-75: for my taste here (and also in the Discussion) are to many "may". If you do not have sufficient data to support your claims then omit them. Some parts of the manuscript are way too speculative.

L85-94: consider shifting this part up in the introduction. The break from the previous paragraph reads awkward.

L97: I think it would be good to briefly justify the markers used for the study. Mitochondrial 16S rDNA has been used previously for population level studies but histone H3? This is a quite conserved marker that has predominantely been used for deeper phylogenetics, so I was surprised to see that you now employ it for population level/phylogeographic studies. Even more I was suprised that you found more haplotypes in H3 than in 16S. However, a brief justification why you have used H3 in thios study would be in order. Has this marker been used proviously in echinoderms (the citation for the primers you provide is for arthropods), had this study here exploratory nature or what was the reason to use it here? Is H3 single-copy in holothurians? Also, in L154 you report that a 307 bp fragment of H3 was sequenced. Where in the gene is this stretch located, in which exon? Or was the fragment sequenced from an intron? What is the structure of this gene in holothurians/echinoderms?

L139: JC as best-fit model. Can you also report variations in the different codon positions in H3 (if region is located in an exon)? Have you evaluated different codon models?

L181: genetic isolation of the east coast of Okinawa Island. Teniya is the only population sampled on the east coast (with N=15) and while the pattern you reveal might be real it could also be a simple sampling artefact. Zenpa and Yona are also single haplotype for 16S, and would you not have sampled the other localities on the west coast you would likely speculate complete separation. I would just be a bit more careful and better highlight that the pattern you see might be different with intermediate populations sampled.

L201-204: too speculative. Either test for this using appropriate methods or omit.

L230: "could be due to a shift to asexual reproduction". Again, speculative, should be tested for. Low genetic diversity might simply be an effect of slow evolution of markers used.

L251-256: Again, too many "may" here. I am not at all convinced that that anthropogenic impacts "may" at least partially explain reduction in genetic diversity because they "may" be overexploited...
Have you tested whether markers with presumably higher variation, such as microsatellites or SNPs, show the same reduction in genetic diversity?
Do you have data about population sizes over several years to justify overexploitation?

·

Basic reporting

The manuscript adheres to the PeerJ standards

Experimental design

Experimental design was appropiate

Validity of the findings

The research was conducted in a sound, valid manner.

Have the sequences been deposited to GenBank?

Additional comments

This article was a pleasure to read and was well set out and analyses were appropriate. I have a few suggestions to help improve the manuscript or to consider.

First, there was nothing in the discussion regarding the finding of the non-significance of Fu's statistic. Clearly there is no indication that bottlenecking or expansion has occurred so this should be commented on. It does affect some of your discussion points.

Second, I am not sure that the asexual reproduction is the most likely explanation of these data. It is certainly one possibility but others should be considered. You do already consider the possibility of over exploitation which could reduce genetic diversity. However, have you considered that what might be occurring is that you have sampled the progeny of a limited number of spawning events? If the areas have been overfished then smaller numbers of individuals may contribute to the low diversity seen. Granted, the low diversity is across the sampled areas but this may provide some part of the low diversity in each region and also potentially provide some explanation for the differences seen among the sampled sites.

Third, I am not wholly convinced that the sequenced markers are the best for determining if asexual reproduction has taken place and also if they are the best for determining recent geneflow. Perhaps microsatellites or snp's (e.g. RADseq) might have provided greater diversity and ability to determine if these sampled areas are currently genetically differentiated and reproducing asexually. Perhaps a caveat can be added to suggest future studies with more variable markers might be worthwhile. Although AFLPs are mentioned there are better marker types available now.

Regard,
Mike Gardner

---

## Round 0.2 · Minor Revisions

· Academic Editor

Minor Revisions

My comments are relatively minor corrections or suggestions for clarification that I expect should be simple to address, hence the decision of Minor Revisions. I look forward to seeing the revised manuscript.

One referee mentions the possibility of the existence of a cryptic species assemblage. I can see his point, and feel that it would strengthen the manuscript to address this explicitly in the revision.

Your response: “In the introduction and discussion, we reported that previous studies using AFLP and microsatellites (Uthicke et al., 1999; Uthicke et al., 2001 and Taquet et al., 2011) showed a low number of polymorphic loci in this species. Therefore, in the future we are planning to use a new method such as RADseq to further investigate this. From our previous study of H. edulis (Soliman et al., 2016) and the present study we have recomended that furthur studies are needed in Okinawa Isaland using advanced techniques. Please see L272-L275.“ is not accurate. In fact you do not mention the appropriateness of the markers in the introduction. I think you should justify the use of these markers.

I suggest you could add at the end of the introduction a paragraph stating the scientific hypothesis to be tested and your expectations, given previous results. You would also need to add to results and discussion a comment related to this. This would much enrich your paper, as it stands, does not poke the reader into considering alternative outcomes.

Please define an abbreviation for the first time a gene is mention in the text and then use the abbreviation only. For instance: in line 122 you define the H3 abbreviation as histone H3 (H3) gene. In line 175, you say “nuclear histone H3 (H3) gene”.

Line 110. Please delete “(= populations)” because you are only referring to locations, as there is at this point no data on populations. Also, in all instances where you are referring to locations but use the designation of populations, replace the word populations by locations, e.g., lines 146, 153, 154, 156, 158, 161 etc.

Line 160. “(e.g. (Selkoe et al., 2014; Shanks, 2009). ”, please remove the parenthesis after e.g.

Line 176. “The low genetic diversity observed could be due to shifts to asexual reproduction in this species.” This sentence needs a reference or be re-written, combining it with the next sequence in the text. Alternatively, shift the sentence to the end of the paragraph, appearing as a logic inference of what was previously said.

I expect that it will be relatively straightforward for you to complete these revisions, and I look forward to seeing your revised manuscript. Please address all the points fully.

---

## Round 0.3 · Minor Revisions

· Academic Editor

Minor Revisions

I am sorry for the delay. After careful consideration I think your manuscript still needs some minor adjustments before being accepted. Namely:

1. The title must be changed to closely match the main findings of the paper, e.g. Extremely low genetic variability within and among locations of the greenfish holothurian Stichopus chloronotus Brandt, 1835 in Okinawa, Japan;

2. Given that only 3 haplotypes were detected, I think that Fu’s Fs test is not appropriate and suggest you delete reference to it in the material and methods section and in the results section.

3. The discussion section needs some rearrangement so that first genetic diversity is discussed and only then haplotype frequency.
I invite you to submit a revised version of the manuscript that addresses these points. Once you submit a revised manuscript, I will likely accept it for publication. Thanks again for your patience.

---

## Round 0.4 · accepted · Accept

· Academic Editor

Accept

The paper looks great. Thank you for making the suggested edits. Congratulations on the work and thanks for choosing PeerJ.